# Epidemiology of the Epstein–Barr Virus in Autoimmune Inflammatory Rheumatic Diseases in Northern Brazil

**DOI:** 10.3390/v14040694

**Published:** 2022-03-27

**Authors:** Samires Avelino de Souza França, Julimar Benedita Gomes de Oliveira Viana, Hilda Carla Azevedo Góes, Ricardo Roberto de Souza Fonseca, Rogério Valois Laurentino, Igor Brasil Costa, Aldemir Branco Oliveira-Filho, Luiz Fernando Almeida Machado

**Affiliations:** 1Biology of Infectious and Parasitic Agents Post-Graduate Program, Federal University of Pará, Belém 66075-110, PA, Brazil; samiresfranca@hotmail.com (S.A.d.S.F.); hca.goes@gmail.com (H.C.A.G.); ricardofonseca285@gmail.com (R.R.d.S.F.); 2Virology Laboratory, Institute of Biological Sciences, Federal University of Pará, Belém 66075-110, PA, Brazil; julimarbeneditagomesdeoliveira@gmail.com (J.B.G.d.O.V.); valois@ufpa.br (R.V.L.); 3Evandro Chagas Institute, Health Ministry of Brazil, Ananindeua 67030-000, PA, Brazil; igorbc2003@yahoo.com.br; 4Study and Research Group on Vulnerable Populations, Institute for Coastal Studies, Federal University of Pará, Bragança 68600-000, PA, Brazil; olivfilho@ufpa.br

**Keywords:** rheumatoid arthritis, autoimmune diseases, systemic lupus erythematosus, Epstein–Barr virus

## Abstract

The present study aimed to describe the seroprevalence infection, Epstein-Barr virus (EBV) genotypes, relate the infection’s profile with the epidemiological and corticotherapy data of patients with Autoimmune inflammatory rheumatic diseases (AIRD). A cross-sectional study was carried out with 139 individuals, 92 with systemic lupus erythematosus (SLE), 27 with rheumatoid arthritis (RA) and 20 with other autoimmune diseases, who were undergoing clinical follow-up in Brazil. Serological tests for the detection of EBV anti-VCA IgM and IgG antibodies, as well as the amplification of a segment of the EBV EBNA-3c gene by conventional PCR were performed to identify the infection and the viral subtype. The Epstein–Barr nuclear antigen 3 (EBNA3C) gene participates of maintenance of viral latency and infected B-lymphocytes immortalization by unclear signaling cascades. The association of active/latent EBV infection with EBV infection profile was assessed by Fisher’s exact test and multiple logistic regression. The seroprevalence of EBV anti-VCA IgG was 100%, while that of anti-VCA IgM was 1.43% (2/139). Active-phase infection was confirmed by the presence of EBV DNA in 40.29% of the population evaluated (56/139), with 45.65% (42/92) in SLE, 25.92% (7/27) in the RA and in 35% (7/20) in other autoimmune diseases. It was observed that individuals with SLE had a higher prevalence of active/lytic EBV infection and that oral corticosteroid therapy at a dose lower than 20 mg/day increased the risk of EBV activity by up to 11 times. Only the presence of EBV-1 was identified. Thus, EBV lytic infection was higher in individuals with SLE when compared to other autoimmune diseases with rheumatologic involvement and the lytic activity of the virus precedes corticosteroid-induced immunosuppression.

## 1. Introduction

Autoimmune inflammatory rheumatic diseases (AIRD) comprise a set of clinical disorders, which have in common the production of autoantibodies against several cellular antigens, such as enzymes, certain ribonucleoproteins and DNA itself [1,2,3,4]. All these diseases involve changes at the connective tissue level, having a low prevalence in the general population, but high rates of morbidity and mortality when diagnosed late or treated inappropriately [4]. Young women are generally more affected and clinical manifestations include mucocutaneous, musculoskeletal symptoms (arthralgia/arthritis/myalgia/myositis), renal, gastrointestinal, respiratory and central nervous system changes [5,6].

Epstein-Barr Virus (EBV) belongs to the family *Herpesviridae*, genus *Lymphocryptovirus* and subfamily *Gammaherpesvirinae* and is currently called *Human Gammaherpesvirus 4* [7]. It is quite prevalent in the world and estimates indicate that about 90% of the world population is infected by this virus. The infection may be asymptomatic during life and at any given time, one third of this percentage will present infective particles in the saliva and the symptoms of infectious mononucleosis [8,9,10]. In vitro, EBV expresses nine latency-associated viral proteins, of which six nuclear proteins (EBNA1, EBNA2, EBNA3A, EBNA3B, EBNA3C and EBNA-LP), three membrane proteins (LMP1, LMP2A and LMP2B) and two small molecules of RNA: EBER-1 and EBER-2 [11].

The association of EBV infection and autoimmune diseases, especially systemic lupus erythematosus (SLE), has been reported in several studies, based on serological and molecular evidence [12,13,14]. In individuals with the disease, there is a faster seroconversion to EBV and a higher viral load value when compared to immunocompetent individuals infected by EBV and without SLE [14].

This association is reinforced by the mimicry of the EBV EBNA1 protein with the Ro lupus autoantigen and by the inability of CD8+ T lymphocytes to control virus-infected B-lymphocytes [15,16,17]. Persistent latent EBV infection, with occasional reactivations, immortalization of B lymphocytes and exacerbated T-cell response are among the main aspects related to the autoimmunity observed in systemic sclerosis, and SLE rheumatoid arthritis and Sjogren’s Syndrome [17,18,19]. The involvement of EBV with autoimmune and oncogenic processes is also related to the *EBNA3C* gene, responsible for in vitro immortalization and in vivo lymphomagenesis of infected B lymphocytes. Gene recombination analyses that replaced the wild *EBNA3C* gene with another encoding a dysfunctional protein revealed loss of this transformative potential of EBV [20].

The present study aims to describe the prevalence of EBV infection, the circulating viral genotypes and the profile of EBV infection (active or latent) and to correlate with the clinical characteristics and corticosteroid therapy performed by individuals with various autoimmune diseases who were treated in the city de Belém, Pará, Northern Brazil.

## 2. Materials and Methods

### 2.1. Type of Study and Ethical Aspects

A cross-sectional study was carried out with 139 individuals diagnosed with autoimmune diseases undergoing clinical and laboratory follow-up at the Jean Bittar Hospital (JBH) Outpatient Clinic, located in Belém, Pará, from June to December 2017. Of these, 92 patients had SLE, 27 had a diagnosis of rheumatoid arthritis (RA) and 20 had other autoimmune diseases (Systemic Sclerosis-4, Sjogren’s Syndrome-3, Psoriasis-3, Dermatomyositis-3, Juvenile Idiopathic Arthritis-2, Still’s Disease-2, Polymyositis-1, Disease Bechet-1, Crest Syndrome-1). All individuals were invited to participate in the research while they were waiting for an outpatient medical consultation with the rheumatologist, which took place twice a week. Those who accepted to participate signed a consent form, and blood was collected.

Demographic data (age, gender, clinical diagnosis, date of diagnosis) and epidemiological data (presented symptoms, medications in use, previous laboratory tests, such as blood count, renal and liver function, C-reactive protein [CRP] and erythrocyte sedimentation speed [ESS]) were obtained by through a pre-tested standardized semi-structured questionnaire and medical records. All individuals who had a confirmed diagnosis of AIRD and who were 18 years of age or older were included in the study. All procedures were performed in accordance with the guidelines and regulatory standards for research involving human subjects of the National Health Council. This study was approved by the Ethics Committee on Human Research of the Institute of Health Sciences, Federal University of Pará, under protocol number 2.174.033. A written informed consent was obtained from all 139 patients for the publication of any potentially identifiable images or data included in this article.

### 2.2. Serological Assays

Peripheral blood samples (5 mL) were collected in a vacuum collection system, containing K_2_EDTA as an anticoagulant and was transported to and processed in the Laboratory of Virology of the Institute of Biological Sciences of Federal University of Pará and at the Instituto Evandro Chagas, located in the state of Pará, Brazil. The blood was centrifuged for 10 min at 1400 *g* (3000 rpm) and the plasma and the cell portion were stored at −20 °C until the moment of use.

Epstein–Barr virus capsid antigen IgM (EBV VCA IgM), EBV capsid antigen IgG (EBV VCA IgG) were detected using SERION ELISA classic (Institut Virion\Serion GmbH, Würzburg, Germany) kits following the manufacturer’s instructions. Serological criteria for interpreting lytic or latent viral activity are shown in Table 1.

### 2.3. Molecular Assays

All biological samples collected were submitted to DNA extraction in plasma and whole blood, using the DNeasy^®^ Blood and Tissue kit (Qiagen, Hilden, Germany), according to the manufacturer’s protocol.

For the extraction of genetic material, 200 µL of whole blood and 50 µL of plasma were used, respectively, which were placed in 1.5 mL tubes containing 20 µL of proteinase K and previously identified. 200 µL of lysis buffer were added to the mixture, which was vortexed for 15 s and incubated at 56 °C in a thermoblock for 10 min. After this incubation, 200 µL of ethanol (96–100%) was added, another 15 s vortexing was performed.

The amplifications were performed based on a mix containing 2.5 µL of 10× Buffer; 0.75 µL MgCl_2+_ (50 mM); 1.0 µL of EBNA3C(R) primer—10 pmol/µL; 1.0 µL of EBNA3C primer (F)—10 pmol/µL; 1.0 µL of dNTP (10 mM), 14.5 µL of water; 0.25 µL of Taq polymerase (5 U/µL) and 4 µL of DNA from each sample. The amplification reactions were performed in the SureCycle 8800 thermocycler—Agilent, Santa Clara—United States, based on the following schedule: 5 min at 94 °C, followed by 45 cycles of 30 s at 94 °C, 45 s at 60 °C and 45 s at 72 °C followed 7 min final extension time at 72 °C. The primers used were EBNA3C (R) and (F) specific for EBV (5′ AGA-AGG-GGA-GCG-TGT-GTT-A 3′ e 5′ CGT-GAT-TTC-TAC-CGG-GAG-TGC 3′ (19).

The identification of gene amplification was performed on a 2% agarose gel stained with fluorophore (SYBR^®^ Safe DNA gel stain—Invitrogen, Life Technologies, Carlsbad, CA, USA) using as a parameter a molecular weight of 100 bp (Ladder Ludwig Biotechnology LTDA—Chain from 100 to 1000 bp, with two additional 1500 bands of 2080 bp, each band with 40 ng, except for the 500 bp band with approximately 90 ng). The discrimination of the type of EBV in question was visualized based on the distance traveled by the fragments amplified in the electrophoretic migration, in relation to the Ladder and negative and positive controls, where for EBV-1 there was a fragment of 153 bp and for EBV-2, 246 bp.

### 2.4. Statistical Analysis

Descriptive statistics of the epidemiological, clinical, laboratory and therapeutic management variables of the population were performed, and the results are presented in absolute frequencies and percentages. To assess the association between the prevalence of EBV in relation to its profile of lytic or latent infection and clinical and laboratory variables, a bivariate analysis was performed using Fisher’s exact test. Next, the variables that in the bivariate analysis showed a value of “*p*” less than or equal to 0.20 were included in the Multiple Logistic Regression analysis, based on the Backward method. Analysis were performed using the Minitab 7.0 statistical software (Minitab Inc., Philadelphia, PA, USA) assuming a significance of 5%.

## 3. Results

The study population (*n* = 139) was characterized by being mostly female (92.09%), from 31 to 46 years old (53.96%), single (42.45%), with at least 8 years of study (51.08%), from the capital Belém (49.64%). Among the most attended clinical diagnoses SLE (66.19%) stood out, followed by RA (19.42%) and other diverse and less frequent AIRD (14.48%), such as Still’s disease, Psoriasis, Dermatomyositis, Systemic Sclerosis, Juvenile Idiopathic Arthritis, Bechet’s Disease, Systemic Sclerosis, Polymyositis, Crest’s Syndrome.

The seroprevalence of EBV anti-VCA IgG antibodies was 100%, while only 2 patients (1.43%) were positive for anti-VCA IgM antibodies, the first being RA and the second diagnosed with PS. Patient #27121 with PS had active EBV infection, confirmed by the identification of the EBNA-3c gene in plasma; while patient #28340, with RA, also presented a serological profile of virus lytic activity, but without gene detection in plasma Possibly the amount of plasma used may have reduced the detection sensitivity, or even a low viral load.

Overall, about 40.3% of patients (56/139) had an EBV lytic activity profile, manifesting positivity for the EBNA3C gene in their plasma samples. On the other hand, approximately 59.7% of those evaluated with a diagnosis of autoimmunity had an EBV viral latency profile. In the individualized assessment of autoimmune diagnoses, EBV lytic infection was seen in 45.65% of patients with SLE (42/92); 25.92% (7/27) in those with RA; and in 35% (7/20) of patients with other AIRD. There was exclusive infection by EBV-1 in the evaluated population.

In the bivariate analysis that included the total study population (*n* = 139) and that compared the profile of EBV infection (lytic activity and latency) with clinical laboratory variables (clinical diagnosis, age group, platelet, lymphocyte, ESS, CRP, symptoms) and therapeutic management (medicines used, daily dosage and treatment time) of patients, there was no statistically significant difference between patients with active or latent EBV (Table 2).

Variables stratified by EBV infection profile (lytic activity and latency) with *p*-value < 0.20 were subjected to multiple logistic regression. It was found that individuals with SLE have twice the risk of manifesting EBV lytic activity compared to patients with RA and other AIRD, and this result is statistically significant. From the same perspective, daily doses of oral corticosteroids lower than 20 mg/day were related to an increase of up to 11 times in the risk of EBV lytic activity among all patients, to the detriment of higher doses (Table 3).

In the bivariate analysis, which compared the profile of EBV infection (lytic activity and latency) with the clinical and laboratory variables of patients with SLE (anti-DNA-ds, complement, age group, platelets, lymphocytes, ESS, CRP and symptoms) and treatment (pulse therapy, medications used, daily dose and duration of therapy), there was also no statistical significance between patients with active and latent EBV (Table 4).

None of the variables selected to compose the multiple logistic regression model was significant, although in the bivariate analysis, the isolated use of immunosuppressants showed a significant difference for viral activity. The multiple regression analysis showed that patients with other AIRD who had an elevated ESS inflammatory marker were 8 times more likely to manifest EBV in the lytic phase than patients with ESS within the normal range (Table 5).

## 4. Discussion

The results of the present study point to the relationship of EBV lytic activity with autoimmune diseases, with emphasis on SLE. Although causality relationships cannot be established based on our methodological design, two currents of thought are noteworthy: the authors who point to the molecular mimicry of some EBV antigens interacting with autoantibodies and exacerbating autoimmune disease and on the other hand, authors who defend that the successive reactivations of EBV would be conditioned to pathological imbalances in the immune system of the sick host by SLE, RA or other AIRD [17,18,21,22,23,24,25,26,27].

In our analysis patients with SLE were twice as likely to have EBV in plasma (Table 3—OR 2.5126, *p* < 0,05), characterizing a lytic profile of viral activity, when compared to patients without SLE, but diagnosed with other autoimmune diseases in our study. Confirming this finding, Li et al. [28] found that patients with SLE are up to 4 times more likely to have active EBV in plasma compared to healthy controls; reinforcing that the association between EBV and SLE exists and can also be explained by the prevalence of anti-VCA (IgG, IgA and IgM), anti-EBNA1 (IgA) and anti-EA (IgG, IgA, IgM) antibodies in these patients.

Our results support the theory that EBV would be constantly reactivated in lupus patients leading to immunological disturbances that, in a genetically predisposed individual, may evolve with autoimmune phenomena [2]. The impact of immunosuppressive drug therapy on this process is discussed. However, no studies were found that evaluated treatment-naïve patients, instead, it is observed that drug therapy is always distributed into categories (types of medication and accumulated dose/day), as presented in our analysis. We did not observe a statistically significant difference between the combined use and the accumulated doses of the drugs, related to the lytic activity of EBV in RA and AIRD, which unleashes the viral reactivation of the deliberate drug suppression in these diseases [5]. Furthermore, in SLE, patients who used low doses of corticosteroids showed lytic EBV activity in plasma at the expense of patients with higher doses, reinforcing our result that viral activation in this disease is independent of high doses of corticosteroids and approaches the reactivation cycle of EBV from the pathogenesis of autoimmunity [16].

In contrast with the present study, where the frequency of anti-VCA IgM was low (1.43%); Larsen et al. [29] and Draborg et al. [30] identified high titers of lytic phase antibodies (Anti-EA and Anti-VCA IgA and IgM), discussing the host’s attempt to control the widespread lytic infection in B and epithelial cells that would culminate in the production of various antibody isotypes [26]. It is reinforced that, in our analysis, the identification of the viral genome showed the presence of the virus in lytic phase, regardless of the detection of anti-VCA IgM.

Fattal et al. [31] and Westergaard et al. [32] studied the response to anti-EBNA1 IgA, IgM and IgG antibodies of individuals with autoimmune disease, finding normal, similar, or even reduced titers compared to healthy controls, pointing out that the humoral immune response is dysregulated against EBV infection is conditioned to the lytic phase of viral infection among patients with autoimmune disease, which corroborates our findings regarding the high frequency of viral DNA visualized in the plasma of individuals.

Interestingly, the immune disturbance common to RA seems to be related to the latent EBV infection pattern. This information is compatible with our findings, which revealed a viral latency profile in 74.07% of individuals with RA [21,22,23,24,25,26,27,28]. According to Westergaard et al. [32] RA patients have high titers of anti-EBNA1 IgM, IgG and IgA antibodies compared to healthy controls and SLE patients, further reinforcing this observed latency profile. Furthermore, patients with RA have a high EBV viral load detectable in PBMC confirming the impact of viral latency on the pathophysiology of RA [33,34].

According to the results of the present study, EBV manifests itself in the lytic phase more predominantly among patients with SLE. Although drug immunosuppression is discussed as a bias for viral reactivation observed in these patients, our results showed that patients who did milder and earlier therapeutic regimens manifested EBV lytic activity, placing the virus as one of the potential etiological agents of SLE [15,16,28,30].

In RA and AIRD, a viral latency profile was observed regardless of the therapeutic regimen (isolated/associated) or drug dose. These different forms of presentation of the infection between different autoimmune diseases reinforce the impact of gene expression of the different stages of viral latency on the genesis of RA and AIRD, and reaffirm the immunological weakness observed among patients with SLE, in which there is greater lack of control of the EBV lytic infection [24,32].

In patients with other AIRD (*n* = 20), high levels of ESS in the blood were related to a significantly increased risk (OR: 8.33) of lytic EBV infection. ESS is a laboratory marker used in the diagnosis of various clinical conditions and in the assessment of their severity, being a nonspecific test in the documentation of an inflammatory, infectious or neoplastic process, but useful to infer intensity and response to therapy, especially in rheumatologic conditions [35]. Therefore, there is evidence that the chronic inflammatory process inherent to autoimmunity conditions also contributes to EBV reactivation, especially among other AIRD [33,34].

Unfortunately, our work was limited to the exclusive analysis of patients with established autoimmune disease. Thus, we point out a limitation of our analysis, which did not compare the profile of EBV infection in lupus patients with a healthy control group. Another limitation of this study concerns the exclusive serological analysis of anti-VCA IgG and IgM. Information related to the profile of anti-EBNA1 IgM and IgG antibodies and others of early lytic phase, such as anti-EA IgM, could complement and reinforce the results [27,28,29,30].

It is possible that our overall sample universe (*n* = 139) was too small to be able to build comparative relationships of significance in the bivariate analyses. This was more evident when autoimmune diseases were separated according to specific diagnoses: SLE (*n* = 92), RA (*n* = 27) and other AIRD (*n* = 20), analyzed individually and it was found that clinical and laboratory variables and therapy were not statistically significant. However, it is noteworthy that in the multivariate analysis of the total number of patients in the study (*n* = 139), significance was detected for EBV lytic activity among patients with SLE and those who used daily corticosteroids at lower doses, reinforcing that the phase lysis of EBV infection is one of the complex factors related to the pathogenesis of SLE.

## 5. Conclusions

Finally, EBV infection appears to be related to autoimmune diseases, as this group of patients manifests free viral particles in their plasma samples. The lack of control between cellular and humoral immunity in an attempt to contain EBV even in latency is one of the main triggers of the successive reactivations of the infection seen in SLE, and this repetitive exchange between latency stage and viral activity is listed as the main suspect in the pathophysiology of autoimmunity associated with viral infection.

## Figures and Tables

**Table 1 viruses-14-00694-t001:** Laboratory criteria for classifying the EBV infection profile.

Anti-VCA * IgM	Anti-VCA IgG	Infection Profile
+	+	Active
+	+/−	Active
−	+	Latent
−	+	Latent

* virus capsid antigen.

**Table 2 viruses-14-00694-t002:** Clinical, laboratory and therapeutic characteristics according to the profile of EBV infection in autoimmune diseases.

	EBV Active		EBV Latent		*p*-Value
Parameters	*n* (%)	CI * (95%)	*n* (%)	CI * (95%)	
Lupus					
Positive	42 (75.00)	61.63; 85.81	50 (60.24)	49.90; 70.83	0.1049
Negative	14 (25.00)	14.39; 38.37	33 (39.76)	29.17; 51.10	
Rheumatoid arthritis					
Positive	7 (12.50)	5.18; 24.07	20 (24.10)	65.27; 84.62	0.1398
Negative	49 (87.50)	75.93; 94.82	63 (75.90)	15.38; 34.73	
Others AIRD					
Positive	7 (12.50)	5.18; 24.07	13 (15.66)	8.61; 25.29	0.7835
Negative	49 (87.50)	75.93; 94.82	70 (84.34)	74.71; 91.39	
Age (Years)					
Up to 38	36 (64.29)	50.36; 76.64	44 (53.01)	41.74; 64.07	0.2526
>38	20 (35.71)	23.36; 49.64	39 (46.99)	35.93; 58.26	
Platelets					
Normal	48 (85.71)	73.18; 93.62	76 (91.57)	83.39; 96.54	0.4168
Altered	8 (14.29)	6.38; 26.22	7 (8.43)	3.46; 16.61	
Lymphocytes					
Normal	47 (83.93)	71.67; 92.38	68 (81.93)	71.95; 89.52	0.9383
Altered	9 (16.07)	7.62; 28.33	15 (18.07)	10.48; 28.05	
ESS **					
Normal	37 (66.07)	52.19; 78.19	65 (78.31)	69.91; 86.61	0.1597
Altered	19 (33.93)	21.81; 47.81	18 (21.69)	13.39; 32.09	
CRP ^†^					
Normal	30 (53.57)	39.74; 67.01	54 (65.06)	53.81; 75.20	0.2372
Altered	26 (46.43)	32.99; 60.26	29 (34.94)	24.80; 46.19	
Symptoms					
Asymptomatic	10 (17.86)	8.91; 30.40	17 (20.48)	12.41; 30.76	0.8688
Symptomatic	46 (82.14)	69.60; 91.09	66 (79.52)	69.24; 87.59	
Drugs					
Immunosuppressive medication only	40 (48.19)	37.08; 59.44	33 (58.93)	44.98; 71.90	0.2845
Immunosuppressive medication + corticoid	43 (51.81)	40.56; 62.92	23 (41.07)	28.10; 55.02	
Immunosuppressive medication per day					
Up to 400 mg	36 (64.29)	50.36; 76.64	54 (65.06)	53.81; 75.20	1.0000
>400 mg	20 (35.71)	23.36; 49.64	29 (34.94)	24.80; 46.19	
Corticoid per day					
Up to 20 mg	75(90.36)	81.89; 95.75	1 (1.79)	0.05; 9.55	0.1351
>20 mg	8(9.64)	4.25; 18.11	55 (98.21)	90.45; 99.95	
Diagnosis period					
Up to 5 years	34 (60.71)	46.75; 73.50	54 (65.06)	53.81; 75.20	0.7323
>5 years	22 (39.29)	26.50; 53.25	29 (34.94)	24.80; 46.19	

Confidence Interval *, Erythrocyte Sedimentation Speed **, C-reactive protein ^†^.

**Table 3 viruses-14-00694-t003:** Adjusted analysis of variables associated with the EBV lytic infection profile of patients with autoimmune disease (*n* = 139).

Parameters	OR *	CI ** 95%	*p*-Value
Lupus	2.5126	1.1524; 5.4782	0.0205
Corticoid dose per day > 20 mg	11.0099	1.2716; 95.3253	0.0294

Odds Ratio *, Confidence Interval **.

**Table 4 viruses-14-00694-t004:** Clinical, laboratory and therapeutic characteristics according to the profile of EBV infection in lupus patients (*n* = 92).

	EBV Active		EBV Latent		*p*-Value
Parameters	*n* (%)	CI * (95%)	*n* (%)	CI * (95%)	
Anti-DNA-ds					
Non reagent	31 (73.81)	57.96; 86.14	39 (78.00)	64.04; 88.47	0.8227
Reagent	11 (26.19)	13.86; 42.04	11 (22.00)	11.53; 35.96	
Complements					
Normal	32 (76.19)	60.55; 87.95	34 (68.00)	53.30; 80.48	0.5243
Altered	10 (23.81)	12.05; 39.45	16 (32.00)	19.52; 46.70	
SLEDAI **					
Active	19 (45.24)	29.85; 61.33	20 (40.00)	26.41; 54.82	0.7682
Non active	23 (54.76)	38.67; 70.15	30 (60.00)	45.18; 73.59	
Proteinuria					
Present	27 (64.29)	48.03; 78.45	23 (46.00)	31.81; 60.68	0.1226
Absent	15 (35.71)	21.55; 51.97	27 (54.00)	39.32; 68.19	
Pulse Therapy					
Present	21 (50.00)	34.19; 65.81	31 (62.00)	47.17; 75.35	0.3444
Absent	21 (50.00)	34.19; 65.81	19 (38.00)	24.65; 52.83	
Age (Years)					
Up to 38	28 (66.67)	50.45; 80.43	36 (72.00)	57.51; 83.77	0.7441
>38	14 (33.33)	19.57; 49.55	14 (28.00)	16.23; 42.49	
Platelets					
Normal	36 (85.71)	71.46; 94.57	46 (92.00)	80.77; 97.78	0.5296
Altered	6 (14.29)	5.43; 28.54	4 (8.00)	2.22; 19.73	
Lymphocytes					
Normal	33 (78.57)	63.19; 89.70	43 (86.00)	72.76; 94.06	0.5090
Altered	9 (21.43)	10.30; 36.81	7 (14.00)	5.94; 27.24	
ESS ***					
Normal	30 (71.43)	55.42; 84.28	41 (82.00)	73.26; 94.18	0.3400
Altered	12 (28.57)	15.72; 44.58	9 (18.00)	5.82; 26.74	
CRP ^†^					
Normal	28 (66.67)	50.45; 80.43	40 (80.00)	66.28; 89.97	0.2253
Altered	14 (33.33)	19.57; 49.55	10 (20.00)	10.03; 33.72	
Symptoms					
Asymptomatic	8 (19.05)	88.60; 31.12	11 (22.00)	11.53; 35.96	0.9283
Symptomatic	34 (80.95)	65.88; 91.40	66 (79.52)	64.04; 88.47	
Drugs					
Immunosuppressive medication only	10 (20.00)	10.03; 33.72	28 (66.67)	50.45; 80.43	0.0467
Immunosuppressive medication + corticoid	40 (80.00)	66.28; 89.97	14 (33.33)	19.57; 49.55	
Immunosuppressive medication per day					
Up to 400 mg	23 (54.76)	38.67; 70.15	25 (50.00)	35.53; 64.47	0.8057
>400 mg	19 (45.24)	29.85; 61.33	25 (50.00)	35.53; 64.47	
Corticoid per day					
Up to 20 mg	8 (16.00)	7.17; 29.11	41 (97.62)	87.43; 99.94	0.0660
>20 mg	42 (84.00)	70.89; 92.83	1 (2.38)	0.06; 12.57	
Diagnosis period					
Up to 5 years	23 (54.76)	38.67; 70.15	31 (62.00)	47.17; 75.35	0.6242
>5 years	19 (45.24)	29.85; 61.33	19 (38.00)	24.65; 52.83	

Confidence Interval *, Systemic Lupus Erythematous Disease Activity Index **, Erythrocyte Sedimentation Speed ***, C-reactive protein ^†^.

**Table 5 viruses-14-00694-t005:** Adjusted analysis of variables associated with the EBV lytic infection profile of patients with other AIRD (*n* = 20).

Parameters	OR *	CI ** 95%	*p*-Value
High ESS ^†^—EBV active	8.3330	1.0343; 67.1384	0.0464

Odds Ratio *, Confidence Interval **, Erythrocyte Sedimentation Speed ^†^.

## Data Availability

Not applicable.

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
