# Peer review of "Epidemiology of the Epstein–Barr Virus in Autoimmune Inflammatory Rheumatic Diseases in Northern Brazil"

_viruses, 2022, doi:10.3390/v14040694_

Round 1

Reviewer 1 Report

as stated previously, the study design could have been stronger with a control group comprised of individuals not diagnosed with autoimmune disease and treated with steroids to address the fact that steroids will generally result in increased EBV.

Reviewer 2 Report

Minor revision

Line 132: The Authors write untruth; parenthesis (19) was not corrected on [19], correct.

Reviewer 3 Report

Line 112: Table 1.

The last two rows are repeated in the newly resubmitted document.

  • - + Latent
  • - + Latent

This manuscript is a resubmission of an earlier submission. The following is a list of the peer review reports and author responses from that submission.

Round 1

Reviewer 1 Report

This paper is looking at EBV reactivation in patients diagnosed with autoimmune diseases. The only significant finding is the increased presence of active viral replication (lytic) in individuals treated with immunosuppressive drugs. I have 2 concerns regarding the study design and conclusions of this paper.

  1. There should be a control of individuals without autoimmune diseases who are treated with steroids. One would expect that due to the high EBV seropositivity in the population, they too would experience increased lytic reactivation when treated with immunosuppressive drugs. It is difficult to fund an association with autoimmune disease without this control.
  2. The discussion presents the findings in a light that suggests that they are more significant than they are. For example page 7, line 220 it says: "Patients with SLE are twice as likely to have EBV in plasma, characterizing a lytic profile of viral activity, when compared to patients without SLE, but diagnosed with other 221 autoimmune diseases." Where is the analysis supporting this? Yes, the number is higher but the p-value suggests that this is not a significant difference. The authors should review the discussion to make sure that all claims are supported by statistics.
  3. The first paragraph of the paper seems to be from another paper completely. It is about hepatitis prevalence. There is no clear connection to the rest of the paper.

Author Response

REQUESTS - REVIEWER 1:

This paper is looking at EBV reactivation in patients diagnosed with autoimmune diseases. The only significant finding is the increased presence of active viral replication (lytic) in individuals treated with immunosuppressive drugs. I have 2 concerns regarding the study design and conclusions of this paper.

  1. There should be a control of individuals without autoimmune diseases who are treated with steroids. One would expect that due to the high EBV seropositivity in the population, they too would experience increased lytic reactivation when treated with immunosuppressive drugs. It is difficult to fund an association with autoimmune disease without this control.

Author’s response: The authors agree with the suggestion of the second reviewer, who emphasizes the importance of a control group of individuals who use steroids but do not have autoimmune disease. In fact, the control group of this nature would discriminate whether EBV reactivations would be related to the use of medication or to the consequences of the autoimmune disease. However, the design of our study aimed to compare patients between the autoimmune disease groups, since the entire sample was using some corticosteroid therapy, either at low dose (<20mg/day) or high dose (>20mg/d ) and there were no treatment virgins. Thus, patients were separated according to their clinical diagnoses into 3 groups (SLE, RA and other autoimmune diseases) and then compared with reference to daily corticosteroid dose (<20mg/day and >20mg/day). Our results showed that patients with SLE-type autoimmune disease manifested 2-fold higher EBV lytic activity when compared to patients with RA and other diseases. In this sense, SLE seems to be a risk factor for the visualization of the lytic activity of the virus, to the detriment of other related autoimmune diseases. Future analyzes may include a control group to detect the influence of corticosteroid therapy on this process.

  1. The discussion presents the findings in a light that suggests that they are more significant than they are. For example page 7, line 220 it says: "Patients with SLE are twice as likely to have EBV in plasma, characterizing a lytic profile of viral activity, when compared to patients without SLE, but diagnosed with other 221 autoimmune diseases." Where is the analysis supporting this? Yes, the number is higher but the p-value suggests that this is not a significant difference. The authors should review the discussion to make sure that all claims are supported by statistics.

Author’s response: We add to the discussion a return to table 3, which shows the result that SLE had an OR 2.5126 for EBV lytic activity compared to RA and other autoimmune diseases, with statistical significance p=0.0205. Based on this result, the authors argue that the clinical diagnosis of SLE was related to a double chance of finding EBV in lytic activity among patients, when compared to RA and other autoimmune diseases. The result brings to light the impact of SLE pathophysiology on the viral cycle, or the impact of viral activity on SLE disease, but this analysis requires a prospective study design. The text in the discussion was changed to:

Lines 218-220: In our analysis patients with SLE were twice as likely to have EBV in plasma (Table 3 - OR 2.5126, p<0,05)…

  1. The first paragraph of the paper seems to be from another paper completely. It is about hepatitis prevalence. There is no clear connection to the rest of the paper.

Author’s response: We agree with the reviewer and apologize for the misunderstanding. The first paragraph has been deleted.

Reviewer 2 Report

Comments and Suggestions for Authors:

The present study refers to describe the prevalence of EBV infection, the circulating viral genotypes, and the profile of EBV infection (active or latent) and to correlate with the clinical characteristics and corticosteroid therapy performed by individuals with various autoimmune diseases who were treated in the city de Belém, Pará, Northern Brazil.

The Epstein-Barr virus is one of the most common viruses in humans and at the same time the mysterious cause of many diseases, including cancerous. In the last time the association of EBV infection and autoimmune diseases, especially systemic lupus erythematosus (SLE), has been reported in several studies, based on serological and molecular evidence. Hence, I consider the originality of this manuscript is high.

The research concept is interesting, but some changes and corrections should be made to better understand the manuscript.

Major revision:

  1. A poorly structured introduction. Why do the Authors write in the introduction about hepatitis viruses A, B, and C. Manuscript concern Epstein Barr virus. In addition, the literature cited in this point (Lines: 41-50), reference 1-4 are non-specific for hepatitis only for rheumatic diseases. Correct needed.
  2.  Both the abstract (Line 26) and the introduction must contain information on the studied EBNA3 / EBNA3C (Epstein-Barr nuclear antigen 3) gene, the functions of which are crucial for the persistence and latency of EBV, along with other genes of this family.

Minor revision:

  1. Expand the abbreviation (AIIRD) in the abstract (Line 23 ).
  2. Changed bracket „(19)” to „[19]” (Line 134)
  3. What is HJB? (Line: 156), DRAIS, ESR
  4. Explain all abbreviations used in the manuscript (DRAIS, ESR, CRP-protein, and so on...).

Author Response

REQUESTS - REVIEWER 2:

The present study refers to describe the prevalence of EBV infection, the circulating viral genotypes, and the profile of EBV infection (active or latent) and to correlate with the clinical characteristics and corticosteroid therapy performed by individuals with various autoimmune diseases who were treated in the city de Belém, Pará, Northern Brazil.

The Epstein-Barr virus is one of the most common viruses in humans and at the same time the mysterious cause of many diseases, including cancerous. In the last time the association of EBV infection and autoimmune diseases, especially systemic lupus erythematosus (SLE), has been reported in several studies, based on serological and molecular evidence. Hence, I consider the originality of this manuscript is high.

The research concept is interesting, but some changes and corrections should be made to better understand the manuscript.

Major revision:

  1. A poorly structured introduction. Why do the Authors write in the introduction about hepatitis viruses A, B, and C. Manuscript concern Epstein Barr virus. In addition, the literature cited in this point (Lines: 41-50), reference 1-4 are non-specific for hepatitis only for rheumatic diseases. Correct needed.

Author’s response:We appreciate the observation and we deleted the first paragraph.

  1. Both the abstract (Line 26) and the introduction must contain information on the studied EBNA3 / EBNA3C (Epstein-Barr nuclear antigen 3) gene, the functions of which are crucial for the persistence and latency of EBV, along with other genes of this family.

Author’s response: We agree with the observation and added paragraphs that cite the EBV genes, especially EBNA3C. Text included in the manuscript:

Abstract:

Line 8: The Epstein-Barr nuclear antigen 3 (EBNA3C) gene participates of maintenance of viral latency and infected B lymphocytes immortalization by unclear signaling cascades.

Introduction:

Lines 15-18: In vitro, EBV expresses nine latency-associated viral proteins, of which six nuclear proteins (EBNA1, EBNA2, EBNA3A, EBNA3B, EBNA3C and EBNA-LP), three membrane proteins (LMP1, LMP2A and LMP2B) and two small molecules of RNA: EBER-1 and EBER-2 [11];

Lines 29-33: The involvement of EBV with autoimmune and oncogenic processes is also related to the EBNA3C gene, responsible for in vitro immortalization and in vivo lymphomagenesis of infected B lymphocytes. Gene recombination analyses that replaced the wild EBNA3C gene with another encoding a dysfunctional protein revealed loss of this transformative potential of EBV [19].

Minor revision:

  1. Expand the abbreviation (AIIRD) in the abstract (Line 23 ).

Author’s response: The abbreviation was spelled out and also changed from AIIRD to AIRD.

  1. Changed bracket „(19)” to „[19]” (Line 134)

Author’s response: It was corrected.

  1. What is HJB? (Line: 156), DRAIS, ESR

Author’s response: The acronym HJB has been corrected, it refers to the Regional Hospital where the research data was collected: Hospital Jean Bittar (first appearance on line 79-80).

  1. Explain all abbreviations used in the manuscript (DRAIS, ESR, CRP-protein, and so on...).

Author’s response: The other acronyms were also expanded throughout the text in their first appearance, including in the tables. In the second paragraph of Materials and Methods, the acronyms of CRP and ESS were placed for the first time.

Reviewer 3 Report

In this manuscript, Franca, and colleagues aim to study the prevalence of EBV infection, the viral genotype, latent and lytic virus infection, and to relate them with the epidemiological and corticotherapy data of patients with various autoimmune diseases who were treated in Northern Brazil. The cross-sectional study was carried out with 139 individuals, the seroprevalence of RBV anti-VCA IgG was 100% and anti-VCA IgM was 1.43%. The lytic phase of viral DNA was confirmed in 40.29% are all of genotype-1 only. Further, the individual clinical and therapy profiles correlated with EBV infection.

CRITIQUES: (1 -4)

1. The main title (line2-3) is misleading. Specifically with the words ‘expression of EBNA3 gene

The expression of EBNA3 has not been examined in this study and the EBNA3 gene is expressed as mRNA and protein during the latent phase of the virus. Therefore, EBNA3 is not a suitable gene to distinguish latent-lytic phases of EBV infection. Further, the authors in this manuscript performed PCR only using Taq polymerase that helps to EBNA3 from DNA source (that is genomic EBV DNA). To test mRNA expression of EBV one has to do reverse-transcriptase PCR. Here the author did EBNA3 PCR citing Sample et al., 199 paper for genotype identification only and they succeeded in the identification of EBV genotype 1.

Therefore, the title needs to be changed from ‘expression of EBNA3 gene’ to ‘genotyping’ or removed entirely, because the term epidemiology includes serology and genotype data.

Or change as below;

Epidemiology of Epstein-Barr virus in autoimmune inflammatory rheumatic diseases in Northern Brazil

2. Remove/modify columns one and two in table 1.

Considering the above-listed comments with EBNA3 PCR that amplify the EBV genomic DNA and that does not inform whether the virus is in the lytic or latent phase, please remove columns one and two in table 1. EBNA3 PCR informs us of EBV genotype 1 or 2.

However, antibodies to EBNA may be used for testing that can help distinguish whether someone is susceptible to EBV infection or has a recent or past infection.

https://www.cdc.gov/epstein-barr/laboratory-testing.html

The EBNA1 IgG ELISA is recommended for the determination of past infections. The authors could rely on VCA ELIS to distinguish latent and lytic phases.

https://www.serion-diagnostics.de/en/products/serion-elisa-classic-antigen/epstein-barr-virus/

3. Table 2-5: please use dot (.) and not a comma (,) for decimal numbering.

Table 4 has many * symbols without explanations

4. Introduction line 42 to 50 is about hepatitis and it has to be removed and replaced with EBV-related text and references.

The World Health Organization (WHO) recognizes that viral hepatitis is a global    42
public health issue; according to a WHO report published in 2017, it has been estimated    43

that 71 million people are infected with one of the four major hepatitis viruses, each of    44

which can cause approximately 1.5 million deaths and significantly affect the lives of    45

millions of individuals globally every year [1,2]. In Brazil, viral hepatitis is an epidemic    46

public health problem and is mainly caused by hepatitis viruses A, B, and C. Hepatitis C    47

virus (HCV) is an enveloped RNA virus that was first identified in 1989 and has an ex-   48

traordinary degree of genetic diversity; each virus can currently be classified into seven    49

different genotypes and more than 80 subtypes [3,4].

Author Response

REQUESTS - REVIEWER 3:

In this manuscript, Franca, and colleagues aim to study the prevalence of EBV infection, the viral genotype, latent and lytic virus infection, and to relate them with the epidemiological and corticotherapy data of patients with various autoimmune diseases who were treated in Northern Brazil. The cross-sectional study was carried out with 139 individuals, the seroprevalence of RBV anti-VCA IgG was 100% and anti-VCA IgM was 1.43%. The lytic phase of viral DNA was confirmed in 40.29% are all of genotype-1 only. Further, the individual clinical and therapy profiles correlated with EBV infection.

  1. The main title (line2-3) is misleading. Specifically with the words ‘expression of EBNA3 gene

The expression of EBNA3 has not been examined in this study and the EBNA3 gene is expressed as mRNA and protein during the latent phase of the virus. Therefore, EBNA3 is not a suitable gene to distinguish latent-lytic phases of EBV infection. Further, the authors in this manuscript performed PCR only using Taq polymerase that helps to EBNA3 from DNA source (that is genomic EBV DNA). To test mRNA expression of EBV one has to do reverse-transcriptase PCR. Here the author did EBNA3 PCR citing Sample et al., 199 paper for genotype identification only and they succeeded in the identification of EBV genotype 1.

Therefore, the title needs to be changed from ‘expression of EBNA3 gene’ to ‘genotyping’ or removed entirely, because the term epidemiology includes serology and genotype data.

Or change as below:

Epidemiology of Epstein-Barr virus in autoimmune inflammatory rheumatic diseases in Northern Brazil

Author’s response: We greatly appreciate the reviewer's remark and have changed the title of the manuscript as suggested.

  1. Remove/modify columns one and two in table 1.

Considering the above-listed comments with EBNA3 PCR that amplify the EBV genomic DNA and that does not inform whether the virus is in the lytic or latent phase, please remove columns one and two in table 1. EBNA3 PCR informs us of EBV genotype 1 or 2.

However, antibodies to EBNA may be used for testing that can help distinguish whether someone is susceptible to EBV infection or has a recent or past infection.

https://www.cdc.gov/epstein-barr/laboratory-testing.html

The EBNA1 IgG ELISA is recommended for the determination of past infections. The authors could rely on VCA ELIS to distinguish latent and lytic phases.

https://www.serion-diagnostics.de/en/products/serion-elisa-classic-antigen/epstein-barr-virus/

Author’s response: columns 1 and 2 of Table 1 were removed. Activity or latency criteria were analyzed according to the profile of anti-VCA IgM and IgG antibodies.

  1. Table 2-5: please use dot (.) and not a comma (,) for decimal numbering.

Author’s response: It was corrected.

  1. Table 4 has many * symbols without explanations

Author’s response: The meaning of the abbreviations was added at the bottom of the table with the corresponding symbols.

  1. Introduction line 42 to 50 is about hepatitis and it has to be removed and replaced with EBV-related text and references.

The World Health Organization (WHO) recognizes that viral hepatitis is a global    42
public health issue; according to a WHO report published in 2017, it has been estimated    43

that 71 million people are infected with one of the four major hepatitis viruses, each of    44

which can cause approximately 1.5 million deaths and significantly affect the lives of    45

millions of individuals globally every year [1,2]. In Brazil, viral hepatitis is an epidemic    46

public health problem and is mainly caused by hepatitis viruses A, B, and C. Hepatitis C    47

virus (HCV) is an enveloped RNA virus that was first identified in 1989 and has an ex-   48

traordinary degree of genetic diversity; each virus can currently be classified into seven    49

different genotypes and more than 80 subtypes [3,4].

Author’s response: It was corrected. We deleted the first paragraph.

Round 2

Reviewer 1 Report

I still believe that the control group of individuals without autoimmune disease on steroids would strengthen this study. In the response to the review, the authors highlighted table 3 which shows a statistically significant increased odds ratio among patients with SLE compared to other patients, however, the steroid treatment had a much larger impact. The authors discuss the limitations of their study and the conclusions that can be drawn sufficiently.

Reviewer 3 Report

The authors have fully addressed the critiques raised in the review reports and improved the overall quality of the manuscript.